# Gas Chromatography with Flame-Ionization Detection-Based Analysis of Sugar Contents in Korean Agricultural Products for Patients with Galactosemia

**DOI:** 10.3390/foods12051104

**Published:** 2023-03-05

**Authors:** Ha-Neul Jeong, Ryeong Ha Kwon, Yuri Kim, Sang-Ho Yoo, Seon Mi Yoo, Chi-Do Wee

**Affiliations:** 1Department of Agro-Food Resources, National Institute of Agricultural Sciences, Rural Development Administration, Wanju 55365, Republic of Korea; 2Department of Nutritional Science and Food Management, Ewha Womans University, Seoul 03760, Republic of Korea; 3Department of Food Science & Biotechnology, and Carbohydrate Bioproduct Research Center, Sejong University, Seoul 05006, Republic of Korea

**Keywords:** galactosemia, galactose content, commercial agro-food, gas chromatography, flame-ionization detection

## Abstract

Patients with galactosemia accumulate galactose in their bodies, requiring a lifelong galactose-restricted diet. Therefore, accurate information on the galactose content in commercial agro-food resources is essential. The HPLC method generally used for sugar analysis has low separation and detection sensitivity. Here, we sought to establish an accurate analytical method for determining the galactose content in commercial agro-food resources. To that aim, we employed gas chromatography with flame-ionization detection to detect trimethylsilyl-oxime (TMSO) sugar derivatives (concentration: ≤0.1 mg/100 g). The galactose content in 107 Korean agro-food resources reflecting intake patterns was then analyzed. The galactose content in steamed barley rice was 5.6 mg/100 g, which was higher than that in steamed non-glutinous and glutinous rice. Moist-type and dry-type sweet potatoes, blanched zucchini, and steamed Kabocha squash had high galactose content (36.0, 12.8, 23.1, and 61.6 mg/100 g, respectively). Therefore, these foods are detrimental to patients with galactosemia. Among fruits, avocado, blueberry, kiwi, golden kiwifruit, and sweet persimmon had galactose contents of ≥10 mg/100 g. Dried persimmon had 132.1 mg/100 g and should therefore be avoided. Mushrooms, meat, and aquatic products showed low galactose content (≤10 mg/100 g), making them safe. These findings will help patients to manage their dietary galactose intake.

## 1. Introduction

Currently, an average of about 40,000 newborns, equivalent to approximately 10% of the total number of deliveries, are born with congenital abnormalities. According to the statistics from 10,000 births over the last 10 years, there has been an approximately threefold increase in the number of newborns with congenital abnormalities. About 300 hereditary metabolic diseases are currently known, and approximately 100 of them appear in the neonatal period. Dietary substrate restriction to reduce the utilization of abnormal metabolic pathways is a necessary treatment for some hereditary metabolic diseases [1]. Galactosemia is a genetic disease in which galactose accumulates in the body due to a deficiency of enzymes galactokinase (GALK), UDP-galactose 4-epimerase (GALE), and galactose-1-phosphate uridylyltransferase (GALT) involved in the metabolic process that converts galactose into glucose. The birth frequency of sick children in Korea calculated based on the results of the newborn screening test conducted in 2005 was reported to be approximately one in 40,000 [2]. If treatment is delayed, galactosemia can cause fatal complications such as mental retardation, spasticity, cataracts, breastfeeding problems, hypoglycemia, stunted growth, jaundice, hepatocellular damage, hemorrhage, and *E. coli* sepsis. However, once treatment is started with rapid diagnosis, lifespan can be normal, and neurodevelopment can be improved [2,3].

The sugars contained in food are either naturally present or added during processing [4,5]. These not only improve flavor and prolong the storage period but also act as an essential energy source in the body. However, patients with galactosemia are deprived of breastfeeding and must follow a strict lifelong galactose-restricted diet [2,3]. Currently, there is not much information on the trace amounts of galactose contained in agro-food resources. In Korea, patients with galactosemia manage their diet based on galactose content information (number of foods: 276), which is provided in precision units by the US Galactosemia Foundation as a standard for managing dietary intake. However, this information is for food produced and consumed in the United States, there is no information for each stage of cooking and processing, and most of the information is outdated (i.e., obtained before 2000). Therefore, it is difficult to apply to Korean patients.

A simple and rapid high-performance liquid chromatography (HPLC) method has recently been adopted in many laboratories and most studies for the quantitative analysis of sugars, including galactose, in food. However, when a refractive index (RI) or evaporative light scattering (ELS) detector is used, the sensitivity and resolution are low, hindering the analysis of samples at low concentrations [6,7]. Therefore, it is difficult to use the data for the management of galactosemia. In addition, carbohydrate and NH_2_ columns used to separate glucose and galactose, and epimer are expensive. Therefore, a more economic and accurate method using other universal instruments is sorely needed.

Gas chromatography/mass spectrometry (GC/MS) equipped with a capillary column is generally used to analyze volatile organic compounds. GC analysis of saccharides with relatively low molecular weight is performed by derivatizing the hydroxyl groups present in their structures. Among them, trimethylsilyl (TMS) and trimethylsilyl-oxime (TMSO) derivatization methods are widely used [8,9,10]. In fact, in the case of capillary GC analysis equipped with a flame ionization detector (FID), the detector’s sensitivity is higher than that of HPLC analysis. Accordingly, it is possible to analyze low-concentration saccharides contained in the sample at the ppm level, and the peak resolution is superior to that of HPLC [6,7,8,9,10]. Therefore, GC/MS is advantageous because it provides relatively high qualitative and quantitative properties in samples containing complex substrates and economic efficiency due to the long-term use of the column. For patients with galactosemia, GC/FID analysis of low-concentration galactose contained in agro-food resources is rare [11,12]. Almost all available data on galactose content in foods are provided by the US Galactose Foundation. However, these data only reflect the galactose content in raw food and do not consider processing and type of intake. Furthermore, the data are very limited. Therefore, patients with galactosemia are at risk of complications because of the lack of information.

Here, our aim was to analyze the content of sugars, including galactose, in agro-food resources produced and consumed in Korea to provide nutritional guidance for patients with galactosemia.

## 2. Materials and Methods

### 2.1. Sample Selection

Agricultural samples are the top foods consumed according to the Korea Rural Economic Institute’s food balance sheet, national reports, and statistical data. We screened for foods high in sugar requested for analysis by patients with galactosemia, guardians, and clinical nutritionists, and consumed by multiple persons during a meal survey using the 24-h recall method for children with galactosemia and general children [13]. A total of 107 species were selected as follows: 17 cereals and their products, seven potatoes and starches, 11 beans, five mushrooms, 16 vegetables, 25 fruits, 17 meats and eggs, and nine seafoods. All samples were purchased, then freeze-dried and pulverized for use in the experiments.

### 2.2. Standards and Reagents

Standards (glucose, galactose, fructose, sucrose, lactose, and maltose), hexamethyldisilazane (HMDS), trifluoro acetic acid (TFA), pyridine, and hydroxylamine hydrochloride were purchased from Sigma-Aldrich Co. (St. Louis, MO, USA). Other reagent-grade chemicals and HPLC-grade solvents (ethanol, methanol, and water) were obtained from Thermo Fisher Scientific (Waltham, MA, USA).

The sugar standard solution was prepared by weighing 50 mg of each of the standards in a 50 mL volumetric flask and dissolving it in 50% methanol. The solutions were stored in a refrigerator at 4 °C until further use in the experiments. Phenyl β-d-glucopyranoside (Sigma-Aldrich) was used as an internal standard.

### 2.3. Sample Preparation

After lyophilization, 2 g of the homogenized sample was weighed, placed in 10 mL of 60% ethanol, kept in an 85 °C water bath for 30 min, and then cooled. The extract was centrifuged at 3600 rpm for 20 min. The supernatant was added to 0.5 mL of 10% lead acetate followed by centrifugation at 3600 rpm for 20 min to remove the proteins. The supernatant was added to 10% oxalic acid to precipitate the surplus lead acetate. The collected extract was filtered through a Whatman No.1 filter paper. The sample was then quantitatively transferred to a 50 mL volumetric flask and diluted to mark with 60% ethanol. One mL aliquots of each extract and 0.5 mL of the internal standard were added and concentrated using N2 gas.

### 2.4. TMSO Derivatization

TMSO derivatization was conducted as previously described with slight modifications [14]. Briefly, 500 µL of hydroxylamine hydrochloride solution dissolved in pyridine at a concentration of 25 mg/mL was added to the concentrated dry residue, stirred in a water bath at 75 °C for 30 min, and cooled at room temperature for 30 min. Then, 450 μL of HMDS and 50 μL of TFA were added and mixed thoroughly. The reaction mixture was subsequently derivatized in a water bath at 85 °C for 60 min and cooled at room temperature for 60 min. For GC analysis, the sample was filtered using a 0.22 μm PVDF membrane filter, placed in GC vials, and used for analysis.

### 2.5. GC/FID Analysis Conditions

GC separation of six free sugars was performed using an Agilent 7890A GC system equipped with an automatic sampler and an FID following a previously described method with slight modifications [15,16,17]. Two-microliter samples were injected. Free sugars were separated using a non-polar HP-5 column (30 m × 0.25 mm × 0.25 μm; Agilent Technologies). The injection and detector temperatures were set at 280 °C and 300 °C, respectively. N_2_ was used as a carrier gas, and samples were separated at a rate of 0.9 mL/min. Table 1 shows the heating conditions of the oven. All samples were performed in duplicate.

## 3. Results and Discussion

### 3.1. Resolution of GC for TMSO Sugar Derivatives

The chromatogram of the GC/FID system for six free sugar standard solutions and an internal standard solution is shown in Figure 1. All sugars were separated without interfering peaks in the order of fructose, galactose, glucose, sucrose, maltose, and lactose. The retention time (RT) values were 14.566, 16.203, 16.699, 26.978, 28.602, 29.563, and 24.03 min for fructose, galactose, glucose, sucrose, lactose, maltose, and the internal standard, respectively. All were detected within 30 min.

TMSO derivatization was performed by preparing six free sugar standard solutions at concentrations of 0.1, 0.2, 0.4, 1, 2, 5, 10, and 20 mg/100 g; a calibration curve for the standard material was prepared to validate the analytical method. The calibration curve’s determination coefficients (R^2^) were 0.9995, 0.9992, 0.9992, 0.9992, 0.9989, and 0.9986 for fructose, galactose, glucose, sucrose, lactose, and maltose, respectively. All six free sugars showed excellent linearity (Figure 2).

The limit of detection (LOD) and limit of quantification (LOQ) were calculated to determine the minimum detectable and quantifiable concentration of fructose, glucose, galactose, sucrose, maltose, and lactose and thus verify the accuracy of the analysis. In each standard chromatogram, the LOD was the value obtained by multiplying the standard deviation of 10 noise peak area values in the blank by three and adding the average, while the LOQ was the value obtained by adding the average to the value multiplied by 10, as in the method previously described [14]. The LOD was 0.2137, 0.3445, 0.2972, 0.1216, 0.0361, and 0.0523 mg/100 g for fructose, galactose, glucose, sucrose, lactose, and maltose, respectively; the LOQ was 0.6477, 1.0439, 0.9005, 0.3685, 0.1093, and 0.1585 mg/100 g, respectively. These LOD and LOQ values are lower than 5–20 mg/100 g of LOD achieved by HPLC/ELSD or HPLC/RID according to previous studies [6,7], demonstrating that the GC/FID method after TMSO derivatization allows detection at lower concentrations (Table 2).

### 3.2. Analysis of Free Sugars in Agro-Food Resources

Separation and quantification of fructose, galactose, glucose, sucrose, maltose, and lactose were simultaneously performed on 107 agro-food resource samples using the GC/FID analysis method established in this study. Table 3 shows the results of quantifying galactose in precise units for patients with galactosemia, the purpose of this study.

The galactose content of cereals, including rice cakes, and their products ranged from 0 to 12.6 mg/100 g. Galactose was not detected in steamed non-glutinous rice and glutinous rice, whereas steamed barley rice contained a relatively high level of galactose (5.6 mg/100 g), which is harmful to patients with galactosemia. The galactose content in boiled somyeon was 6.8 mg/100 g, which was higher than that in spaghetti (3.20 mg/100 g) or buckwheat (0.5 mg/100 g) noodles. The galactose content in steamed corn was 5.0 mg/100 g, which was relatively high. Sirutteock had a high galactose content of 12.6 mg/100 g due to the influence of gomul.

The galactose content in steamed and roasted superior potatoes was 4.6 mg/100 g and 5.5 mg/100 g, respectively. The galactose content in steamed Japanese and Garnet sweet potatoes was 12.8 mg/100 g and 36.0 mg/100 g, respectively. Therefore, these are very harmful foods for patients with galactosemia. However, galactose was not detected in starch prepared from sweet potato, potato, or corn.

Beans, nuts, and seeds generally showed a low galactose content of less than 3 mg/100 g. However, boiled chestnuts had a high content (16.8), making them dangerous foods for patients with galactosemia. Boiled red beans showed a low galactose content of 0.2 mg/100 g, but red bean paste showed a high galactose content of 58.3 mg/100 g due to the influence of additives.

One study [18] stated that free galactose is affected by cultivar differences. The storage time of fruits and vegetables and galactose content in vegetables in this study also showed a wide distribution range of 0–61.6 mg/100 g. The galactose content in frequently and heavily consumed foods such as napa cabbage, ugeoji, green onion, and lettuce showed a low value of less than 5 mg/100 g. Iceberg lettuce and perilla leaf showed high galactose contents of 29.1 mg/100 g and 10.5 mg/100 g, respectively. Blanched zucchini and steamed Kabocha squash showed high galactose contents of 23.1 mg/100 g and 61.6 mg/100 g, respectively, making them very harmful to patients.

The galactose content in mushrooms was relatively low, ranging from 0.2 to 3.5 mg/100 g. The galactose content in fruits showed a wide distribution, ranging from 0 to 132.1 mg/100 g. The galactose content in avocado, blueberry, kiwifruit, and golden kiwifruit was higher than 10 mg/100 g. Sweet persimmon showed a slightly higher galactose content of 13.4 mg/100 g, while dried persimmon showed an exceptionally high galactose content of 132.1 mg/100 g, making it a food that patients should avoid.

Both meat and aquatic products were safe food groups for patients with galactosemia, as they showed a relatively low galactose content of less than 10 mg/100 g. The content in fried eggs was relatively high, ranging from 6.6 to 8.5 mg/100 g. In the case of meat, the galactose content in blanched meat was lower than that in grilled meat. Therefore, free sugar may be dissociated and released during steaming. Galactose, a water-soluble substance, is often released from food into the cooking water during wet cooking. Therefore, removing the used cooking water can effectively reduce the galactose content. However, since wet cooking using heat can significantly reduce the palatability of food, it is necessary to use an appropriate wet cooking method.

## 4. Conclusions

Patients with galactosemia find it difficult to manage their diets while eliminating galactose, which is contained in trace amounts in agricultural food resources. In this study, 107 commercially available agri-food resources frequently consumed by Koreans were analyzed for galactose content according to intake type using a GC system equipped with an FID detector. Galactose, which is present in food in small amounts, was analyzed at the level of 0.01 mg/100 g, and the contents were classified according to food groups.

The sensitivity of the method was much higher than that of the widely used HPLC analytical method. The galactose content in steamed pearl barley, garnet sweet potato, blanched zucchini, and steamed Kabocha squash was very high. Thus, these are harmful foods for patients with galactosemia. Among fruits, the galactose content in avocado, blueberry, kiwifruit, golden kiwifruit, and sweet persimmon was relatively high, while dried persimmon showed a remarkably high galactose content and must be avoided by patients. Mushrooms, meat, and aquatic products showed low galactose content, making them safe for patients with galactosemia. Our findings will help patients in Korea to manage their diets. However, access to processed foods has increased greatly in recent years, posing a risk to patients with galactosemia. Therefore, the galactose content in processed foods also needs to be accurately determined.

## Figures and Tables

**Figure 1 foods-12-01104-f001:**
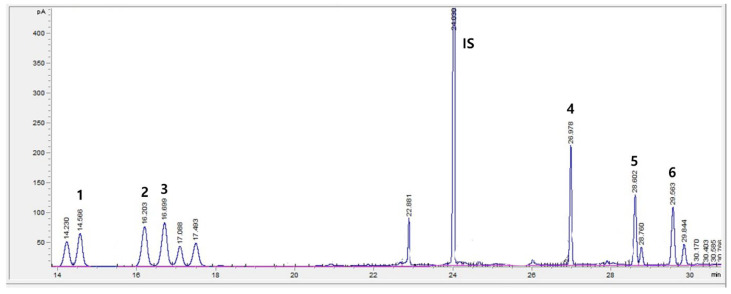
Chromatogram of the separation of six major free sugars by gas chromatography/flame ionization detector (GC/FID). 1 = fructose, 2 = galactose, 3 = glucose, 4 = sucrose, 5 = lactose, 6 = maltose, IS = Internal Standard.

**Figure 2 foods-12-01104-f002:**
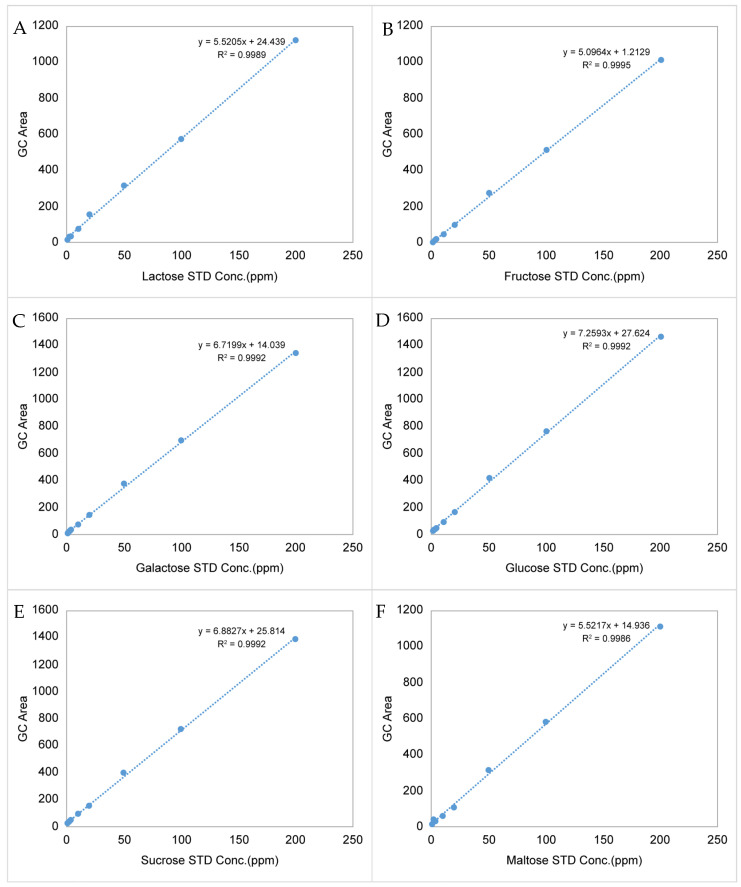
Linearity and determination coefficients (R^2^) of the calibration curves for six major free sugars by GC/FID. (**A**) = lactose, (**B**) = fructose, (**C**) = galactose, (**D**) = glucose, (**E**) = sucrose, (**F**) = maltose, IS = Internal Standard.

**Table 1 foods-12-01104-t001:** GC/FID analysis conditions for accurate analysis of galactose content.

Item	Condition				
Instrument	GC-FID (Agilent 7890A)				
Column	Agilent HP-5 (30 m × 0.25 mm × 0.25 μm)				
Gradient		Rate	Value	Hold Time	Run Time
	Initial		180 °C	5 min	5 min
	Ramp 1	1 °C/min	195 °C	0 min	20 min
	Ramp 2	20 °C/min	280 °C	10 min	34.25 min
	Ramp 3	50 °C/min	295 °C	3 min	34.75 min
	Ramp 4	30 °C/min	180 °C	3 min	44.383 min
Flow rate	0.9 mL/min				
Inlet temp.	280 °C				
Detector	Flame Ionization Detector (FID)				
Injection vol.	2 μL				
Split ratio	10:1				

**Table 2 foods-12-01104-t002:** LOD and LOQ of GC/FID for six major free sugar standards.

Free Sugar	Standard Deviation (σ)	Slope of the Calibration Curve (S)	Limit of Detection (LOD) (mg/100 g)	Limit of Quantitation (LOQ (mg/100 g)
Fructose	3.3011	5.0964	0.2137	0.6477
Galactose	7.0147	6.7199	0.3445	1.0439
Glucose	6.5372	7.2594	0.2972	0.9005
Sucrose	2.5364	6.8827	0.1216	3.68520
Lactose	0.6032	5.5205	0.0361	0.1093
Maltose	0.8429	5.3172	0.0523	0.1585

**Table 3 foods-12-01104-t003:** Galactose content in different agricultural food resources according to intake type.

Food Group	No.	Sample	Galactose Content (mg/100 g FW)			No.	Sample	Galactose Content (mg/100 g FW)		
Cereals and their products	1	flour (cake flour)	4.73	±	0.11	10	corn, steamed	5.04	±	0.40
	2	flour (bread flour)	3.99	±	0.28	11	Baek-seolgi	1.94	±	0.46
	3	non-glutinous rice, white rice		nd		12	Songpyeon, sesame	0.61	±	0.87
	4	glutinous rice, steamed		nd		13	Jeungpyeon	3.94	±	0.50
	5	pearl barley, steamed	5.57	±	0.34	14	wheat Tteok-bokki rice cake (boiled)	3.49	±	0.55
	6	brown rice, steamed	2.00	±	0.12	15	rice cake soup (boiled)		nd	
	7	spaghetti, boiled	3.21	±	0.03	16	Sirutteock	12.63	±	0.91
	8	somyeon, boiled	6.76	±	5.22	17	Injeolmi		nd	
	9	buckwheat noodles, boiled	0.53	±	0.24					
Potatoes and starches	1	superior, steamed	4.63	±	0.03	5	sweet potato starch		nd	
	2	superior, roasted	5.51	±	0.22	6	potato starch		nd	
	3	Japanese sweet potato, steamed	12.75	±	0.29	7	corn starch		nd	
	4	Garnet sweet potato, steamed	36.01	±	0.38					
Beans, nuts, and seeds	1	soybean, boiled	1.26	±	0.04	7	almond, roasted		nd	
	2	green flesh black bean, boiled	2.07	±	0.06	8	chestnut, boiled	16.79	±	0.18
	3	red bean, boiled	0.16	±	0.09	9	peanut, roasted	3.63	±	0.49
	4	red bean paste	58.34	±	2.58	10	sesame, white sesame, roasted		nd	
	5	soybean sprout, blanched	1.13	±	0.01	11	perilla, roasted		nd	
	6	soybean sprout, steamed	0.98	±	0.10					
Mushrooms	1	oyster mushroom, blanched	0.19	±	0.27	4	enoki mushroom, blanched	0.08	±	0.12
	2	shiitake mushroom, blanched	3.15	±	0.38	5	matsutake, grilled	3.54	±	0.06
	3	white button mushroom, grilled	0.57	±	0.12					
Vegetables	1	napa cabbage, boiled	2.18	±	0.26	9	perilla leaf	10.48	±	0.25
	2	napa cabbage, ugeoji, boiled		nd		10	spinach, blanched	1.89	±	0.16
	3	garlic, parboiled	1.27	±	0.18	11	zucchini, blanched	23.09	±	1.16
	4	green onion, blanched	4.74	±	0.05	12	broccoli, blanched	5.89	±	0.50
	5	carrot, blanched	11.36	±	0.51	13	paprika, roasted	10.61	±	0.46
	6	tomato	6.36	±	0.02	14	cucumber	4.30	±	0.09
	7	lettuce	3.96	±	0.03	15	eggplant, blanched	5.27	±	0.18
	8	iceberg lettuce	29.08	±	1.74	16	Kabocha squash, steamed	61.56	±	4.41
Fruits	1	tangerine (citrus unshiu)		nd		14	Cherry	6.90	±	0.66
	2	tangerine (dekopon)	3.88	±	0.52	15	sweet persimmon	13.41	±	3.37
	3	tangerine (setoka)	6.30	±	1.40	16	dried persimmon	132.09	±	3.71
	4	apple (Fuji)	5.42	±	0.11	17	ripe persimmon	7.06	±	0.48
	5	aori apple		nd		18	plum	7.47	±	0.25
	6	banana	6.47	±	0.45	19	kiwifruit	21.05	±	0.99
	7	peach (nectarine)	3.99	±	0.51	20	Golden kiwifruit	11.45	±	0.49
	8	peach (white peach)	3.12	±	0.19	21	grapefruit		nd	
	9	pear	8.52	±	0.17	22	watermelon	4.95	±	0.25
	10	strawberry (sulhyang)	2.46	±	0.15	23	lemon	6.67	±	1.19
	11	avocado	10.05	±	0.21	24	pineapple	2.45	±	0.19
	12	blueberries	13.43	±	0.70	25	green grape		nd	
	13	mango	0.63	±	0.89					
Meat and eggs	1	beef, sirloin, grilled	5.76	±	0.17	10	whole egg, boiled	3.64	±	0.24
	2	beef, beef plate, boiled	2.53	±	0.09	11	whole egg, fried, hard-boiled	6.64	±	0.46
	3	pork, pork belly, grilled	6.33	±	0.06	12	whole egg, fried, soft-boiled	7.21	±	0.80
	4	pork, sirloin, boiled	3.12	±	0.34	13	egg yolk, boiled	5.27	±	0.14
	5	chicken, boiled	0.27	±	0.38	14	egg yolk, fried, hard-boiled	8.48	±	0.45
	6	chicken, leg, grilled	0.27	±	0.00	15	egg yolk, fried, soft-boiled	8.12	±	0.51
	7	chicken, breast, grilled		nd		16	scrambled egg, roasted	5.55	±	0.16
	8	Tea-smoked duck (grilled)	6.73	±	0.53	17	quail egg, boiled	2.87	±	0.28
	9	pork liver, boiled	5.97	±	0.53					
Aquatic products	1	mackerel, grilled	3.24	±	0.16	6	pollack, dried pollack, dried	2.04	±	0.16
	2	hairtail, grilled	3.46	±	0.79	7	mussel, boiled	0.66	±	0.02
	3	squid, blanched	0.70	±	0.01	8	clam, boiled	1.25	±	0.04
	4	stir-fried anchovy, stir-fried baby anchovy		nd		9	abalone, boiled	5.71	±	0.72
	5	anchovy broth		nd						

## Data Availability

No new data were created or analyzed in this study. Data sharing is not applicable to this article.

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
