# Peer review of "Gas Chromatography with Flame-Ionization Detection-Based Analysis of Sugar Contents in Korean Agricultural Products for Patients with Galactosemia"

_foods, 2023, doi:10.3390/foods12051104_

Round 1

Reviewer 1 Report

  • Title: As the title and research aim (see line 85) stated sugar contents, Table 3 (see line 230) should present not only the galactose content but also the contents of fructose, galactose, glucose, lactose, and maltose.
  • Line 73-79: These sentences need reference(s).
  • Line 109-110: Why was phenyl beta-D-glucopyraniside used as an internal standard?
  • Line 120: Internal standard detail such as the amount or concentration is missing.
  • Line 181-182, 239-240: Information regarding the LOD and LOQ both in HPLC/ELSD and HPLC/RID (from literature) is necessary.
  • Line 184 (Table 2): Why was the LOQ of sucrose the highest?
  • Line 230 (Tabe 3): Were the content of other free sugars (fructose, glucose, etc) observed? The data might add valuable information to the manuscript.
  • Line 219: Why the dried persimmon had so high galactose content?

Author Response

Thank you for your good comments.

Line 73-79: These sentences need reference(s).

-> Please see the attachment.

Line 109-110: Why was phenyl beta-D-glucopyraniside used as an internal standard?

-> This is because phenyl beta-D-glucopyraniside is c7h14o6, which is very similar in structure to glucose but is completely different and is very suitable as an internal standard to check whether sugars are well detected.

Line 120: Internal standard detail such as the amount or concentration is missing.

-> Please see the attachment.

Line 181-182, 239-240: Information regarding the LOD and LOQ both in HPLC/ELSD and HPLC/RID (from literature) is necessary.

-> Please see the attachment.

Line 184 (Table 2): Why was the LOQ of sucrose the highest?

-> This is because sucrose is a disaccharide, so its molecule is large and the sensitivity of the FLD detector is good.

Line 230 (Tabe 3): Were the content of other free sugars (fructose, glucose, etc) observed? The data might add valuable information to the manuscript.

-> In order to confirm that the monosaccharide disaccharide is properly separated and galactose is clearly detected, an analysis method was established along with other saccharides.
Since the purpose of this study is to provide information on the content of free galactose contained in food for patients with galactose metabolic disorders, only the content of galactose was quantitatively produced.

Line 219: Why the dried persimmon had so high galactose content?

-> The cause of the minute difference in galactose content in fruits and vegetables is a task to be solved in the future, and the exact cause is not currently known.

Reviewer 2 Report

This work is interesting and useful, providing nutritional guidance for patients with galactosemia. The authors give a very good overview on the galactose content in commercial agro-food resources, and the findings will help patients to manage their dietary galactose intake. They have defined the purpose of the work very well, demonstrate knowledge of the subject and justify undertaking their research topic. However, the following are the minor comments.

Line 20: Please write “trimethylsilyl-oxime (TMSO)” instead of “TMSO”

Line 42: Please write as: galactokinase (GALK), UDP-galactose 4-epimerase (GALE) and galactose-1-phosphate uridylyltransferase (GALT).

Line 134: “with an automatic sampler and a FID” instead of “with an automatic sampler and an FID “

Lines 164, 169: correct is “determination coefficients (R2)” not “correlation coefficients (R2)”

Please add the information if the results are in agreement/disagreement with those from literature.

In a study (Jim, 2002) was state that the free galactose is affected by cultivar difference, and storage time of fruits and vegetables. I recommend to take in account these aspects in future research.

Jim, V. J. W. (2002). Analysis of free galactose contents during cold storage of four apple cultivars, in thermally treated apples and green beans, and in clear apple juices produced using different enzymatic aids (Doctoral dissertation, University of British Columbia).

Author Response

Thank you for your good comments.

It has been revised in the manuscript according to the reviewer's comments.